# Expanded Newborn Screening in Italy Using Tandem Mass Spectrometry: Two Years of National Experience

**DOI:** 10.3390/ijns8030047

**Published:** 2022-08-09

**Authors:** Margherita Ruoppolo, Sabrina Malvagia, Sara Boenzi, Carla Carducci, Carlo Dionisi-Vici, Francesca Teofoli, Alberto Burlina, Antonio Angeloni, Tommaso Aronica, Andrea Bordugo, Ines Bucci, Marta Camilot, Maria Teresa Carbone, Roberta Cardinali, Claudia Carducci, Michela Cassanello, Cinzia Castana, Chiara Cazzorla, Renzo Ciatti, Simona Ferrari, Giulia Frisso, Silvia Funghini, Francesca Furlan, Serena Gasperini, Vincenza Gragnaniello, Chiara Guzzetti, Giancarlo La Marca, Luisa La Spina, Tania Lorè, Concetta Meli, MariaAnna Messina, Amelia Morrone, Francesca Nardecchia, Rita Ortolano, Giancarlo Parenti, Enza Pavanello, Damiana Pieragostino, Sara Pillai, Francesco Porta, Francesca Righetti, Claudia Rossi, Valentina Rovelli, Alessandro Salina, Laura Santoro, Pina Sauro, Maria Cristina Schiaffino, Simonetta Simonetti, Monica Vincenzi, Elisabetta Tarsi, Anna Paola Uccheddu

**Affiliations:** 1Department of Molecular Medicine and Medical Biotechnology, University of Naples Federico II, 80131 Naples, Italy; 2CEINGE Biotecnologie Avanzate Scarl, 80131 Naples, Italy; 3Newborn Screening, Clinical Chemistry and Pharmacology Lab, Meyer Children’s University Hospital, 50139 Florence, Italy; 4Division of Metabolic Disease, Bambino Gesù Childrens Hospital IRCCS, 00165 Rome, Italy; 5Department of Experimental Medicine, Sapienza University of Rome, 00161 Rome, Italy; 6Department of Mother and Child, The Regional Center for Neonatal Screening, Diagnosis and Treatment of Inherited Congenital Metabolic and Endocrinological Diseases, AOUI, 37126 Verona, Italy; 7Division of Inherited Metabolic Diseases, University Hospital of Padova, 35128 Padova, Italy; 8Dipartimento di Medicina Sperimentale, Sapienza University of Rome, 00161 Rome, Italy; 9Ospedale Civico ARNAS PA, 90127 Palermo, Italy; 10Inherited Metabolic Disease Unit, Pediatric Department, AOUI, 37126 Verona, Italy; 11Center for Advanced Studies and Technology (CAST) and Department of Medicine and Aging Science, “G. d’Annunzio” University of Chieti-Pescara, 66100 Chieti, Italy; 12UOS Malattie Metaboliche e Rare, AORN Santobono, 80131 Naples, Italy; 13U.O.S.D. Screening Neonatale e Patologia Clinica AOU Policlinico Consorziale Ospedale Pediatrico Giovanni XXII Bari, 70121 Bari, Italy; 14LABSIEM (Laboratory for the Study of Inborn Errors of Metabolism), Pediatric Clinic, IRCCS Istituto Giannina Gaslini, 16147 Genoa, Italy; 15Centro Screening Neonatale Regione Marche, U.O.C. Neuropsichiatria Infantile—A.O. Ospedali Riuniti Marche Nord, 61032 Fano, Italy; 16UO Genetica Medica, IRCCS Azienda Ospedaliero-Universitaria di Bologna, 40138 Bologna, Italy; 17Fondazione IRCCS Ca’ Granda Ospedale Maggiore Policlinico, Pediatria Alta Intensità di Cura, 20122 Milan, Italy; 18Pediatric Department, San Gerardo Hospital, 20900 Monza, Italy; 19SSD Endocrinologia Pediatrica e Centro Screening Neonatale, Ospedale Pediatrico Microcitemico “A. Cao”, 09121 Cagliari, Italy; 20Laboratorio Screening Neonatale—Clinica Pediatrica AOU Policlinico “G. Rodolico-San Marco”, 95123 Catania, Italy; 21Laboratory of Molecular Biology of Neurometabolic Diseases, Neuroscience Department, Meyer Children’s University Hospital, 50139 Florence, Italy; 22Dipartimento di Neuroscienze Umane—Unità di Neuropsichiatria Infantile Università Roma Sapienza, 00161 Rome, Italy; 23UO Pediatria, IRCCS Azienda Ospedaliero-Universitaria di Bologna, 40138 Bologna, Italy; 24Dipartimento di Scienze Mediche Traslazionali Università degli Studi di Napoli Federico II, 80131 Naples, Italy; 25SS Screening Prenatale e Neonatale, SC Biochimica Clinica, AOU Città della Salute e della Scienza di Torino, 10126 Torino, Italy; 26Center for Advanced Studies and Technology (CAST) and Department of Innovative Technologies in Medicine and Dentistry, “G. d’Annunzio” University of Chieti-Pescara, 66100 Chieti, Italy; 27SC Pediatria-Malattie Metaboliche, Ospedale Infantile Regina Margherita AOU Città della Salute e della Scienza di Torino, 10126 Torino, Italy; 28Centro Laboratoristico Regionale di Riferimento Screening Neonatale e Malattie Endocrino-Metaboliche UO Pediatria IRCCS Azienda Ospedaliero-Universitaria di Bologna, 40138 Bologna, Italy; 29Center for Advanced Studies and Technology (CAST) and Department of Psychological, Health and Territory Sciences, “G. d’Annunzio” University of Chieti-Pescara, 66100 Chieti, Italy; 30Clinical Department of Pediatrics, San Paolo Hospital, ASST Santi Paolo e Carlo, University of Milan, 20142 Milano, Italy; 31Pediatric Clinic, IRCCS Istituto Giannina Gaslini, 16147 Genoa, Italy

**Keywords:** aminoacidemias, fatty acid oxidation disorders, inborn errors of metabolism, newborn screening, organic acidemias, urea cycle defects, tandem mass spectrometry

## Abstract

Newborn screening (NBS) for inborn errors of metabolism is one of the most advanced tools for secondary prevention in medicine, as it allows early diagnosis and prompt treatment initiation. The expanded newborn screening was introduced in Italy between 2016 and 2017 (Law 167/2016; DM 13 October 2016; DPCM 12-1-2017). A total of 1,586,578 infants born in Italy were screened between January 2017 and December 2020. For this survey, we collected data from 15 Italian screening laboratories, focusing on the metabolic disorders identified by tandem mass spectrometry (MS/MS) based analysis between January 2019 and December 2020. Aminoacidemias were the most common inborn errors in Italy, and an equal percentage was observed in detecting organic acidemias and mitochondrial fatty acids beta-oxidation defects. Second-tier tests are widely used in most laboratories to reduce false positives. For example, second-tier tests for methylmalonic acid and homocysteine considerably improved the screening of CblC without increasing unnecessary recalls. Finally, the newborn screening allowed us to identify conditions that are mainly secondary to a maternal deficiency. We describe the goals reached since the introduction of the screening in Italy by exchanging knowledge and experiences among the laboratories.

## 1. Introduction

Newborn screening (NBS) for inherited metabolic diseases is one of the most advanced tools in precision medicine, as it allows the early diagnosis of genetic diseases so that an effective treatment can be started before the onset of irreversible organ damage.

The introduction of tandem mass spectrometry (MS/MS) in NBS offered the possibility of screening for almost 50 conditions using a single dried blood spot (DBS) [1,2,3].

The first Italian law that organized a NBS national system dates to 1992 (law 104/1992, https://www.gazzettaufficiale.it/eli/id/1992/02/17/092G0108/sg, accessed on 15 April 2022) when the newborn screening for the identification and early treatment of congenital hypothyroidism, phenylketonuria and cystic fibrosis, until then adopted in some regions and not others, was made mandatory for all newborns in Italy.

The development of mass spectrometry led to numerous pilot projects during the last two decades. Reports on regional NBS programs including those based on MS/MS have been published [4,5,6,7,8,9,10,11,12,13].

In order to harmonize the regional situations, a nationwide newborn screening program for inborn errors of metabolism was institutionalized by law between 2016 and 2017 (Law 167/2016, https://www.gazzettaufficiale.it/eli/id/2016/08/31/16G00180/sg, accessed on 15 April 2022; DM 13 October 2016, https://www.gazzettaufficiale.it/eli/id/2016/11/15/16A08059/sg, accessed on 15 April 2022; DPCM 12-1-2017, https://www.gazzettaufficiale.it/eli/id/2017/03/18/17A02015/sg, accessed on 15 April 2022). The DM 13 October 2016 states that the screening program is a system articulated into four main functions (the screening laboratory, the laboratory for confirmatory diagnosis, the clinical centers, and the regional coordination/supervision) that defines the panel of screening conditions, the timing for specimen collection, the screening methodology, the confirmatory tests and the clinical follow up. A periodic review of the list of diseases being screened for is set up by Ministry of Health, in collaboration with other government agencies and organizations (https://www.salute.gov.it/imgs/C_17_pagineAree_1920_0_file.pdf, accessed on 15 April 2022).

The Italian Society for the Study of Inherited Metabolic Diseases and Newborn Screening (SIMMESN) has created a group of experts to collect data about screening activity since 1980 in order to produce an annual summary document (https://www.simmesn.it/it/documenti/rapporti-tecnici-screening-neonatale.html, accessed on 15 April 2022).

A total of 1,586,578 infants born in Italy were screened between January 2017 and December 2020. In this paper, we report on the results from this program between 2019 and 2020, highlighting the progress made since the introduction of the law. We focus on metabolic disorders identified by MS-based analysis. For this survey, we collected data from 15 Italian screening laboratories that cover 97.5% of Italian newborns. To the best of our knowledge, this is the first nationwide survey on NBS in Italy.

## 2. Materials

### 2.1. Data Collection

A total of 1,586,578 infants born in Italy between January 2017 and December 2020 were screened by MS/MS. In 2019, a national web portal was created to facilitate the collection of the NBS results from each screening laboratory and the preparation of the annual national report on behalf of SIMMESN. Here we focus on the results obtained on 806.770 infants screened in Italy between January 2019 and December 2020.

### 2.2. NBS Regional Organization and Coverage

The Italian territory is divided into 20 regions, administratively autonomous territories with defined powers. Each region has identified one reference screening laboratory (except Veneto and Sicily, which have two laboratories) and has the responsibility to organize the NBS system in its territory. Six regions (Basilicata, Friuli, Molise, Trentino Alto Adige, Umbria and Valle D’Aosta) do not have their own screening center and have established interregional agreements with a neighboring region (Basilicata with Puglia, Friuli and Trentino Alto Adige with Veneto, Molise with Lazio, Umbria with Toscana and Valle D’Aosta with Piemonte). Only one region (Calabria) did not perform expanded NBS in 2019 and 2020. However, an agreement has been reached between Campania and Calabria, aimed at implementing expanded NBS for all newborns in Calabria starting in 2021.

In all, 16 laboratories perform NBS tests in Italy (Figure 1). A total of 18 laboratories are involved in biochemical confirmatory testing and 27 in molecular confirmation testing. There are 41 clinical centers responsible for managing and caring for patients.

NBS data for this survey was therefore collected from 15 out of 16 laboratories performing expanded NBS by tandem mass.

The coverage of expanded newborn screening in Italy between 2019–2020 has reached 97.5% of the total neonatal population.

### 2.3. Sample Collection

The recommended time for blood sampling after birth is between 48 and 72 h after birth.

In low-birth-weight infants (<1800 or <2500 gr), blood sample collection is repeated in the first month of life (at approximately 14 days and 30 days) according to specific protocols. In cases treated by parenteral nutrition, a second specimen is collected two to three days after treatment withdrawal. In case of blood transfusion in the first 48 h of life, some regions collect a first sample before the treatment even if it is before 48 h of life and a second one 7 days after treatment discontinuation, other regions repeat the analysis after 15 and 30 days. There is a variability in the definition of gestational weeks associated with preterm infants (<34–36 gestational weeks). In these cases, blood sample collection is repeated in the first month of life (approximately at 14 days and 30 days) according to local protocols.

The time between sampling and analysis depends on the time needed for sample shipment to the screening laboratory. To avoid delays in results, laboratories use courier services, which enable delivery of DBS samples 24–48 h after collection.

The newborn’s personal information, also reported on the Guthrie card, is entered into the laboratory computer system either manually by the laboratory administrative staff, or by the birth center personnel through a web connection interface. In order to avoid manual typing and potential errors resulting from the process, in some centers the unique identifier number of the newborn assigned by the region is used to load data already present in regional databases. Such system makes available the report of the NBS results to the nursing units and eventually to the diagnostic centers. The parents are only informed if an action is required, such as a request for an additional sample.

All the laboratories provide informational materials for parents, available as written brochures or via websites.

### 2.4. Panel of Screened Conditions

Table 1 shows the panel of diseases screened by MS/MS and enclosed in the DM 13 October 2016. They are divided into four main categories: aminoacidemias, organic acidemias, urea cycle defects and fatty acid oxidation disorders. The panel also includes secondary conditions that should be considered in a differential diagnosis since they share biomarkers with some diseases of the main panel.

## 3. Methods

### 3.1. NBS Analysis

The most common method to perform NBS analysis was the less time-consuming unbutylated sample preparation (13/15 laboratories) [14], while the 2 remaining laboratories derivatized the extracted metabolites to butyl esters with HCl in N-butanol [9].

Two laboratories did not use certified kits, while the remaining ones used the commercial kits provided from Perkin Elmer, (PE NeoBase Kit, 10 labs, Pe Neobase-2, 2 labs, Pe NeoGram Kit, 1 lab). Although the majority of laboratories used available certified kits on the market to ensure the accuracy of the test results, few were still using in-house methods in which the validation and performance were done using internal control procedures. Regardless, in all the laboratories, NBS results were consistent with appropriate accuracy and precision.

The most commonly used mass spectrometer was the Waters XEVO TQD (8/15 laboratories) produced by Waters corporation, Milford, MA, USA, then the Perkin Elmer QSight MD produced by Perkin Elmer Massachusetts, USA (4/15) and finally one API 3200, one API 4000 and one API 4500 LC-MS/MS system produced by AB Sciex (Toronto, ON, Canada).

Each screening laboratory has its own cut-off values established on data from a healthy population. The cut-off values are updated regularly. The cut-off values calculated on term infants aged 48–72 h are shown in Appendix A.

### 3.2. Second-Tier Tests

Fourteen laboratories developed second-tier tests and employed them in their workflow to minimize false positives and avoid unnecessary recalls. Table 2 lists the number of screening laboratories performing each second-tier test. The most routinely used second-tier test (13/15 labs) was a fast LC-MS/MS-based method to measure the methylmalonic acid in samples with elevated propionylcarnitine (C3) at the initial screening test. This test is very effective in improving screening performance, due to the poor diagnostic prediction of the primary marker C3 [15].

During the period 2019–2020, there were differences among laboratories on performing the second-tier tests. For a better harmonization of NBS practices in our country, the Italian Society of Neonatal Screening developed training programs to implement the use of that technique.

The mean recall rate for all laboratories was 1.57%, but there was a wide range of screening performance: three laboratories had a recall rate <0.5%, three between 0.5% and 1%, four between 1% and 2% and the five remaining laboratories, >2%.

All screening results were recorded on a software that each laboratory designed to manage NBS samples and data according to their own procedures.

Two recall protocols, high- and low-risk, were adopted by all laboratories based on the type and level of the increased/decreased diagnostic biomarker and on the risk of metabolic decompensation for the suspected diagnosis. In the high-risk protocol, the neonate is immediately referred for clinical evaluation, start of diagnostic confirmatory testing and, when needed, start of specific treatments, including intensive care. In low-risk cases, the neonate is re-tested on an additional DBS collected within 7 days. A flow chart of the NBS process is shown in Figure 2.

### 3.3. Confirmation Testing

Biochemical confirmation of positive cases is made by testing plasma amino acids and acylcarnitines, urinary organic acids, urinary orotic acid, plasma homocysteine and succinylacetone. Plasma vitamin B12 levels are determined to rule out a maternal deficiency in cases with suspected methylmalonic acidurias. Enzymatic assays were mainly performed to confirm disease severity in some fatty acid oxidation disorders (i.e., MCAD and VLCAD deficiency). Molecular analysis was performed in all neonates confirmed by biochemical supplementary tests. Mild hyperphenylaninemias were an exception, since not all clinical centers have genetically characterized positive cases identified by NBS.

Quality controls for newborn screening provided by Centers for Disease Control and Prevention (CDC), Atlanta, Georgia were used by all regional NBS centers. In addition, all 15 Italian NBS laboratories (+16 foreigner laboratories) were involved in the quality control program organized by SIMMESN, providing three surveys per year for phenylalanine evaluation and a yearly Proficiency Testing program (MSITA), consisting of three DBS samples from confirmed patients to evaluate laboratory performance and diagnostic marker interpretation.

Furthermore, 12 of the 15 laboratories also participate in external quality control assurance on blood spots organized by ERNDIM (European Research Network for Inherited disorders of Metabolism) for interpretative acylcarnitine Proficiency Testing and for quantitative evaluation of branched-chain amino acids (including alloisoleucine), phenylalanine, methionine, succinylacetone and homocysteine.

### 3.4. Post-Analytical Tools

Several biomarker ratios are useful for the interpretation of NBS results. Ratios used in the Italian laboratories are reported in Table 1.

Many screening laboratories used Region 4 Stork (R4S), active between 2004–2013 to interpret screening results. Disease ranges were established for each pathological condition and are constantly updated by post analytical tools [16]. A score-condition, calculated using specific disease intervals for all informative analytes, allowed a reduction of false positive rates, improving screening performance [17]. The advanced version of R4S, Collaborative Laboratory Integrated Reports (CLIR) incorporate additional demographic information such as age, birth weight and gender which can be responsible for discrepancies among results collected by several laboratories [18].

## 4. Results and Discussion

A total of 516 infants were confirmed to have metabolic disorders with a percentage of 0.06% of the total newborn population (806,770 live births screened between January 2019 and December 2020). False negative results have not yet been reported to date.

In our experience, aminoacidemias were the most common inborn errors in Italy, representing 52% of positive cases. Equal percentages of patients had organic acidemias and mitochondrial fatty acids beta-oxidation defects (Figure 3).

### 4.1. Aminoacidemias (AA)

Before the introduction of the expanded neonatal screening, the incidence of aminoacidemia diagnosed on clinical bases on Italian territory, excluding PKU and hyperphenylalaninemia, was reported to be 1:36,389 [19].

National data (Table 3) for this group of conditions over the two-year NBS period 2019–2020, identified 270 confirmed cases, with a cumulative incidence of 1:2988, (33.5 patients per 100,000 live births). This category included a large number of PKU/hyperphenylalaninemias (*n* = 257/270, 95% of the total), corresponding to an incidence rate of 1: 3139. The phenotypic distribution observed in Italy was different from the global distribution, where the classical PKU phenotype is dominant [20]. In our population, classical PKU accounted for only 20% of total hyperphenylalaninemias (Figure 3) with an incidence of 1:15,515, a figure close to other European countries and the United States [21], while hyperphenylalaninemias accounted for 76% of aminoacidopathies with an incidence of 1:3935, highlighting the predominance of a milder phenotype in Italy, where most cases do not require dietary or therapeutic intervention. This observation highlights the need for the future development of a standardized system for case definition.

Finally, defects in the regeneration or biosynthesis of the enzyme cofactor tetrahydrobiopterin (BH4) were found to be very rare, as observed worldwide.

Other identified aminoacidopathies (Table 3) included four cases of maple syrup urine disease (MSUD), three cases of classical homocystinuria due to CBS deficiency, three cases of (severe) methylenetetrahydrofolate reductase defect (MTHFR), and one case of tyrosinemia type II.

### 4.2. Organic Acidemias (OA)

A total of 70 cases of OA were identified in Italy during 2019 and 2020 by NBS (Table 3 and Figure 3), most of which were methylmalonic acidemias (n = 38/70, 54%), corresponding to an overall incidence of 1:11,526. This figure is double the estimated incidence in the Italian population (1:21,422) before NBS [19].

Retrospective data relating to methylmalonic acidemias in Italy during the pre-newborn screening period reported an incidence of 1:61,755 [19].

The application of second-tier testing for methylmalonic acid and homocysteine made it possible to lower the C3 cut off (median 4.7 μmol/L), without increasing the burden of unnecessary recalls and allowing the minimization of false negatives that could occur, especially in late-onset CblC patients [22,23]. As a result, the NBS sensitivity increased considerably, improving diagnostic timing of affected newborns. As reported by Kalantari 2022 [24], NBS provides a great benefit of avoiding severe organ damage and diagnostic odyssey in the CblC forms, presenting a wide spectrum of clinical manifestations. Remarkly, CblC showed an incidence of 1: 32,271, making this disease one of the most common inherited metabolic diseases in the Italian population, with one of the highest prevalences in the world. The reported incidence in Portugal was 1:85,000 [22], in Spain and, in the US, 1:100,000 [25,26], while a pilot study in Beijing, China showed the worldwide highest incidence of 1:11,730 [27]. Defining the mutation spectrum in CblC cases was beyond the scope of this paper. At the moment, we cannot define any kind of genotype-phenotype correlation and further functional studies are needed to better define the incidence of this disease.

The remaining 32 other cases of OA (46%) identified by NBS included 14 cases of 2-methylbutyryl-CoA dehydrogenase deficiency, 7 cases of isovaleric academia (IVA), 5 cases of glutaric aciduria type I (GAI), 4 cases of propionic aciduria, 1 case of beta-ketothiolase deficiency and 1 case of 3-hydroxy-3-methylglutaryl-CoA lyase deficiency.

### 4.3. Fatty Acids Beta-Oxidation Defects (FAO Defects)

Eighty-five defects of fatty acid beta-oxidation were identified, corresponding to an incidence of 1:9491 live births (Table 3 and Figure 3). As expected, the highest incidence rate 1:20,686 (39/85 cases among FAO defects, corresponding to 46%) was recorded for MCADD. In the pre-expanded newborn screening era, MCADD was considered a very rare disorder in southern European countries [28], with only 3 symptomatic cases identified in Italy in a 13 year survey [19]. The 2-year NBS study identified 22 cases with VLCADD, which corresponds to an incidence of 1:36,671, which is far more frequent than suggested by a former study on symptomatic patients, which identified only 7 cases in the years 1985–1997 [19]. The striking difference in the number of positive MCADD and VLCADD cases identified by NBS, compared to selective screening in symptomatic patients, suggests that some patients may have died suddenly/unexpectedly without a diagnosis in the pre-screening era or, more likely, that most affected individuals have a milder disease phenotype and are therefore at low risk. Since similarly high incidences have been reported in other countries where these diseases are included in NBS programs, the systematic use of genotyping to detect disease-associated variants, combined with functional enzymatic studies, especially when novel variants are identified [29] ought to be adopted. As stated above, the definition of the mutation spectrum in MCAD and VLCAD cases was beyond the scope of this paper. At the moment, we cannot define any kind of genotype-phenotype correlation and further functional studies are needed to better define the incidence of these diseases.

The other FAO defects identified in the study are listed in Table 3.

### 4.4. Urea Cycle Defects (UCD)

The Italian NBS panel includes citrullinemia Type I, argininosuccinic aciduria and argininemia, while citrin deficiency and HHH syndrome are included in secondary conditions. A total of 28 patients were identified with an overall incidence of 1:28,813. Citrullinemia type I was the most common UCD (14 patients) while argininosuccinic aciduria was diagnosed in 13 cases. One patient was identified with argininemia. Similar to the other disease categories, the overall incidence of UCDs detected by NBS was far more frequent than suggested by a former national study based on selective screening which revealed an incidence of 1:41,506, and which also included ornithine transcarbamylase deficiency (OTCD), the most common UCD [19] (Dionisi-Vici 2002).

### 4.5. Maternal Defects

NBS identified 365 cases of maternal conditions/deficiencies. A total of 328 cases were related to maternal vitamin B12 deficiency with an incidence in our study of 1:4837 newborns, which is more frequent than the incidence of inborn errors of metabolism included in the NBS panel. This result is consistent with other studies which report incidences ranging from 1:2959 in Estonia to 1:4837 in Germany [30,31] and confirm the results of a former Italian regional study in Campania [9]. In the majority of cases, maternal B12 deficiency was due to dietary restrictions and, less frequently, to atrophic gastritis or other pathological conditions.

Other maternal disorders included 15 asymptomatic cases of 3-methylcrotonyl CoA carboxylase deficiency (3MCC), 20 cases of carnitine uptake deficiency (CUD) and 2 mild cases of glutaric aciduria type 1.

## 5. Conclusions

A total of 1,586,578 infants born in Italy were screened between January 2017 and December 2020 after the introduction of the above cited law. For this survey, we collected data on metabolic disorders identified by tandem mass spectrometry (MS/MS) between January 2019 and December 2020. Aminoacidemias were the most common inborn errors in Italy. Equal percentages of organic acidemias and mitochondrial fatty acids beta-oxidation defects were observed.

Second-tier tests were widely applied with good results; methylmalonic acid and homocysteine testing increased considerably the number of identified cases of CblC without increasing the number of unnecessary recalls. A striking finding is the high incidence of CblC defects in the Mediterranean area compared to Northern Europe.

NBS also allows maternal deficiencies which are potentially harmful for the neonate to be identified. Recent data from newborn screening suggest that combining folate supplementation with vitamin B12 supplementation in pregnant women may be useful in reducing the risk of long-term neurologic and intellectual sequelae in children born to women deficient in the vitamin.

The data analysis here presented reflects the benefits of close collaboration among screening and confirmation laboratories, metabolic pediatricians and maternity wards. In Italy, where health care tends to be regionally fragmented, a system for sharing data and evaluating the efficacy of newborn screening programs is essential.

## Figures and Tables

**Figure 1 IJNS-08-00047-f001:**
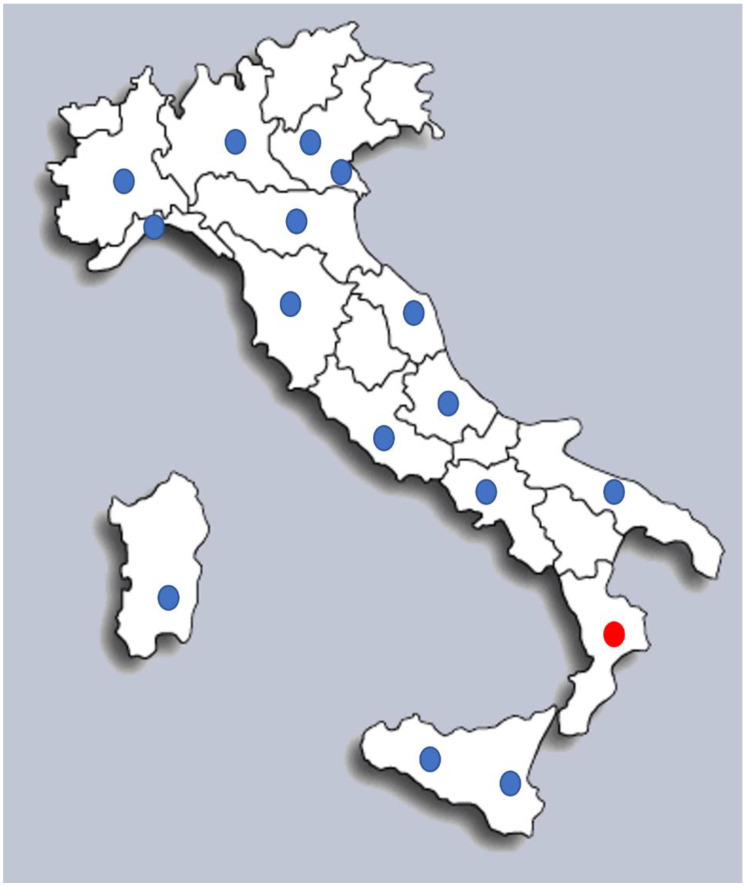
Map of Italian regions. The blue dots represent laboratories performing expanded Newborn Screening in Italy. The red dot marks the only regional laboratory not performing NBS tests by MS/MS in the years 2019–2020. Regions without a dot have established interregional agreements with neighboring region as described in paragraph 2.2 NBS Regional Organization and coverage.

**Figure 2 IJNS-08-00047-f002:**
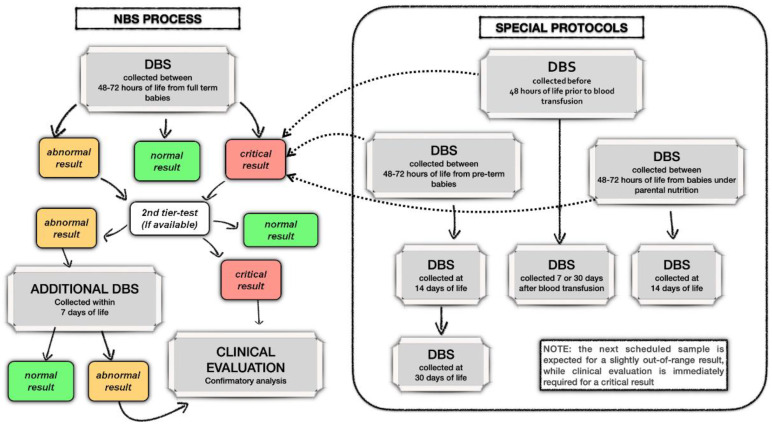
NBS flow chart.

**Figure 3 IJNS-08-00047-f003:**
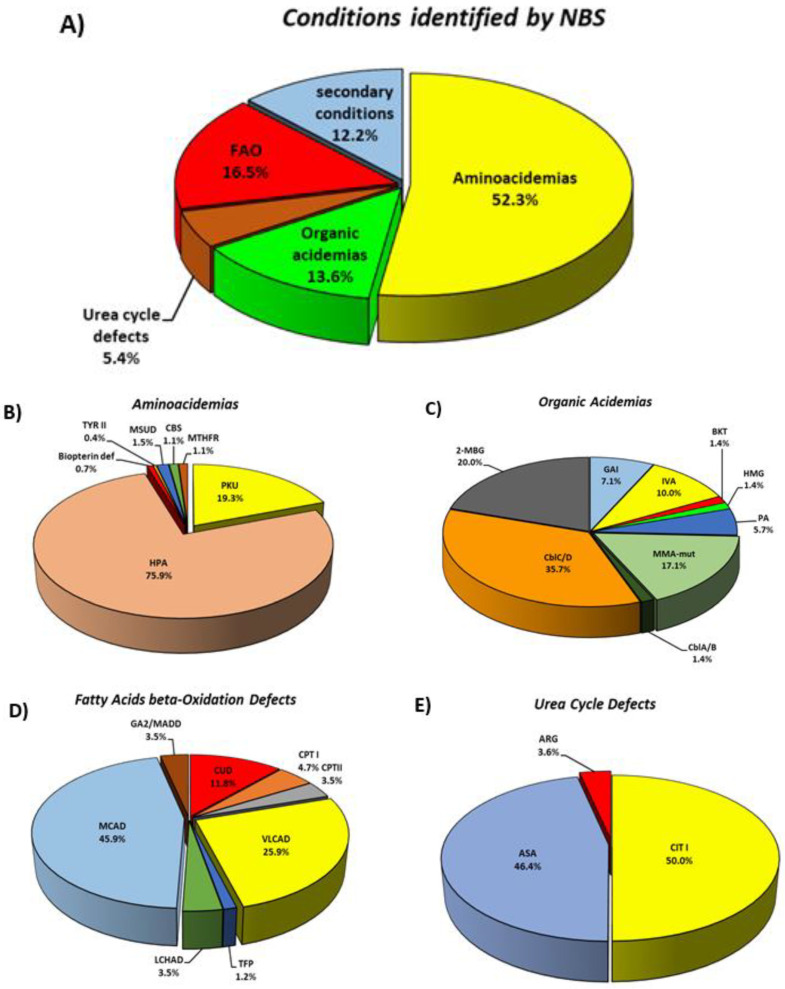
The percentage of different classes of inborn errors of metabolisms in the Italian neonatal population during years 2019–2020, (**A**). The distribution of the different conditions identified by Expanded Newborn Screening, (**B**–**E**).

**Table 1 IJNS-08-00047-t001:** Disorders currently screened by MS/MS-based newborn screening in Italy.

Group	Disorders	OMIM	Biomarkers	Ratios	2nd Tier Test
Aminoacidemias	PKU	261600	Phe	Phe/Tyr	
	HPA	261600			
	Biopterin regeneration/biosynthesis	261630 261640			
	TYR I	276700	Suac (primary marker), Tyr (secondary marker)		Suac with chromatographic separation
	TYR II	276600	Tyr		
	MSUD	248600	Val, Xle	Xle/Ala; Xle/Phe	Ile; Leu; alloIle
	CBS	236200	Met		HCY
	MTHFR	236250	Met ↓		HCY
Organic Acidemias	GAI	231670	C5DC		
	IVA	243500	C5		C5-isomers
	BKT	203750	C5:1, C5OH		
	HMG	246450	C5OH, C6DC		
	PA	606054	C3 (primary marker), C16:1OH (secondary marker)	C3/C2, C3/C16, C3/C0, C4/C3	3-OH-propionic acid; Methylcitric acid; Propionylglycine; Methylmalonic acid; HCY
	MMA-mut	251000			
	CblA/B	251100/251110			
	CblC/D	277400/277410	C3 (primary marker), C16:1OH, Met↓ (secondary markers)	C3/C2, Met/Phe	
	2-MBG	610006	C5		
	MA	606761	C3DC		
	MCD	253270	C5OH		
Urea Cycle Defects	CIT Ia	215700	Cit		
	CIT II	605814	Cit		
	ASA	207900	ArgSucc		
	ARG	207800	Arg		
Fatty Acid Oxidation Disorders	Cud	212140	C0↓, Ctot↓		
	CPT I	255120	C16↓, C18↓	C0/(C16+C18)	
	CACT	212138	C0↓, C16, C18, C18:1		
	CPTII	600650	C16, C18:1, C2↓, C18, C18:2	(C16+C18:1)/C2	
	VLCAD	609575	C14:1, C14, C14:2	C14:1/C2 C14:1/C16	
	TFP	609015	C16OH, C18OH		
	LCHAD	609016	C16OH, C18OH		
	MCAD	201450	C6, C8, C10:1, C10	C6/C8	
	M/SCHAD	231530	C4OH		
	GA2/MADD	231680	C4–C18		
Secondary Conditions	TYR III	276710	Tyr		
	GNMT	606664	Met		
	MAT	250850			
	SAHH	613752			
	3-MGCA	250950	C5OH		
	3-MCC	210200			
	2-M-3-HBD	300438	C5:1 C5OH		
	IBG	611283	C4		Ethylmalonic acid Isobutyrylglycine
	SCAD	201470			

Xle = sum of isomers Ile/Leu/AlloIle/Pro-OH (Isoleucine, Leucine, Alloisoleucine, Hydroxyproline); HCY = Homocysteine; Suac = succinylacetone; ArgSucc = Argininosuccinate; ↓ = decreased.

**Table 2 IJNS-08-00047-t002:** Number of NBS laboratories performing 2nd tier-tests.

2nd Tier-Test	N° Labs
Ile/Leu/Allo	9
Methylmalonic acid	13
2-methylcitric acid	6
Ethylmalonic acid	8
Homocysteine	9
Orotic acid	2
3-OH-propionic acid	3
Propionylglycine	3
C5-isomers	5
Succinylacetone *	2

* Succinylacetone measured in LC-MS/MS to confirm a positive value by FIA-MS/MS screening test.

**Table 3 IJNS-08-00047-t003:** Diagnoses identified by expanded NBS in Italy between 1 January 2019 and 31 December 2020. 806,770 newborns were screened.

Group	Disorders	Total	Incidence
Aminoacidemias	PKU	52	1:15,515
	HPA	205	1:3935
	Biopterin defect in cofactor biosynthesis	1	1:806,770
	Biopterin defect in cofactor regeneration	1	1:806,770
	TYR I	0	-
	TYR II	1	1:806,770
	MSUD	4	1:201,692
	CBS	3	1:268,923
	MTHFR	3	1:268,923
	Total for group	270	1:2988
Organic acidurias	GA I	5	1:161,354
	IVA	7	1:115,253
	BKT	1	1:806,770
	HMG	1	1:806,770
	PA	4	1:201,692
	MMA-MUT	12	1:67,230
	CblA/B	1	1:806,770
	CblC/D	25	1:32,271
	2MBG	14	1:57,626
	Malonic Acidemia	0	-
	MCD	0	-
	Total for group	70	1:11,526
Urea Cycle Defects	CIT I	14	1:57,626
	CIT II	0	-
	ASA	13	1:62,059
	ARG	1	1:806,770
	Total for group	28	1:28,813
Fatty Acid Oxidation Disorders	CUD	10	1:80,677
	CPT I	4	1:20,1692
	CACT	0	-
	CPT II	3	1:268,923
	VLCAD	22	1:36,671
	TFP	1	1:806,770
	LCHAD	3	1:268,923
	MCAD	39	1:20,686
	M/SCHAD	0	-
	GA2/MADD	3	1:268,923
	Total for group	85	1:9491
Secondary Conditions	TYR III	2	1:403,385
	GNMT	0	-
	MAT	13	1:62,059
	SAHH	0	-
	3MGCA	2	1:403,385
	3MCC	23	1:35,077
	2M3HBA	0	-
	IBG	1	1:806,770
	SCAD	22	1:36,671
	CPS	0	-
	Total for group	63	1:12,806
Total of disorders		516	1:1563

AA = Aminoacidemias; OA = Organic Acidurias; UCD = Urea Cycle Disorders; FAO = Fatty Acid Oxidation Disorders.

## Data Availability

The data presented in this study are available on request from the SIMMESN advisory board, through the website https://www.simmesn.it, accessed on 15 April 2022.

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
