# Peer review of "Expanded Newborn Screening in Italy Using Tandem Mass Spectrometry: Two Years of National Experience"

_2409-515X, 2022, doi:10.3390/ijns8030047_

Round 1

Reviewer 1 Report

This paper provides an introduction to NBS programs in Italy with a flow chart of the process and protocols used, and the conditions identified and their incidence.  The description is clear and straightforward and I have only a few questions for the authors:

1. Table 2 lists the number of screening laboratories performing second tier-testing. Many of the 2nd tier-tests are only performed in a few labs, which could increase the number of false-positive cases. Why are 2nd-tier tests not offered more widely by more laboratories?

2. In Figure 2, are there different metabolite marker cutoffs used for newborns undergoing the special protocols?

3. Line 225-228, two labs derivatized the extracted metabolites to butyl esters with HCl in N-butanol and two labs did not use certified kit. Do the sample preparation methods and different kits have an influence on NBS accuracy?

4. Line 248, why is the recall rate very different among labs?

5. Is it correct to conclude from the Table 3 that the total number of NBS true-positive cases confirmed after clinical evaluation is 516, which is approximately 0.06% of total 806,770 newborns screened between 1st January 2019 and 31st December 2020?

6. Are there any false negative cases during this time period?

See typos in Figure 2, TraNsfusion

Line 22, Guthrie card

Line 411-12, This section may be divided by subheadings. It should provide a concise and precise description of the experimental results, their interpretation, as well as the experimental conclusions that can be drawn ?

Author Response

  1. Table 2 lists the number of screening laboratories performing second tier-testing. Many of the 2ndtier-tests are only performed in a few labs, which could increase the number of false-positive cases. Why are 2nd-tier tests not offered more widely by more laboratories?

We thank the reviewer for the interesting observation. When Expanded NBS program was institutionalized by law (2016), few screening laboratories had enough experience in performing the second tier tests, that requires more sophisticate instrumentations for performing it. Nowadays, many screening laboratories include second tier test as routine. The manuscript reported data related to 2019 and 2020 and honestly the performance were not so good. In the 3.2 paragraph, Line 246, pag 7: The following sentence has been added:

“During the period 2019-2020, there were differences among laboratories on performing the second tier tests. For a better harmonization of NBS practices in our Country, the Italian Society of neonatal Screening developed training programs to implement the use of that technique”.

  1. In Figure 2, are there different metabolite marker cutoffs used for newborns undergoing the special protocols?

The survey collected data from 14 labs, requesting the cut-off values calculated on term infants aged 48-72 hours, listed in Table 1S and 2S. As far as we know, many laboratories used different cut-offs for special categories of newborns, calculated according to the age at collection (48-72 hours, 14 and 30 days of life). Currently, we lack of information regarding this aspect.

In the 3.1 paragraph, Line 239, pag 7: The following sentence has been added

The cut-off values calculated on term infants aged 48-72 hours are shown in Table 1S and 2S

Headings of Tables 1S and 2S have been changed in accordance with the above paragraph with the following headings

Table 1S. Cut off calculated on term infants aged 48-72 hours for biomarkers analyzed by a non derivatized method

Table 2S. Cut off calculated on term infants aged 48-72 hours for biomarkers analyzed by a derivatized method

  1. Line 225-228, two labs derivatized the extracted metabolites to butyl esters with HCl in N-butanol and two labs did not use certified kit. Do the sample preparation methods and different kits have an influence on NBS accuracy?

In term of samples handling, preparation and analytical methods all the labs were followed standards and appropriate procedures. The two laboratories that are not using certified kit, performed tests with validated methods according to internal quality system requirements.

In the 3.1 paragraph, Line 233 pag 7:

The following sentence has been added:

“Although the majority of laboratories used available certified kits on the market to ensure

the accuracy of the test results, few were still using in-house methods in which the validation

and performance were done using internal control procedures. Anyway, in all the laboratories, NBS results were consistent with appropriate accuracy and precision.”

  1. Line 248, why is the recall rate very different among labs?

As reported at point 2, in the years 2019-2020, different recall rate was due to the lack in many laboratories of the 2nd tier tests and cut offs setting, not yet performed routinely. A high recall rate has been reported in one laboratory not yet performing second tier tests or using very conservative cut off values with the result of preferring higher sensitivity than specificity. The annual summary reports of our Society is an instrument for a continuous comparison among laboratories in order to homogenize and increase the quality of performance. As a consequence, in 2021, cut off values were updated by many laboratories improving their performances.

  1. Is it correct to conclude from the Table 3 that the total number of NBS true-positive cases confirmed after clinical evaluation is 516, which is approximately 0.06% of total 806,770 newborns screened between 1st January 2019 and 31st December 2020?

We thank the reviewer for the correct observation and valuable comment. We added the information in the manuscript.

At the beginning of section 4 “Results and Discussion”, Line 298 pag 9: The following sentence has been added:

“A total of 516 infants were confirmed to have metabolic disorders with a percentage of 0.06% of the total newborn population (806,770 live births screened between January 2019 and December 2020).”

  1. Are there any false negative cases during this time period?

No false negative results have been reported to date for the years 2019-2020.

Line 300 pag 9: The following sentence has been added:

As far as we know, “False negative results have not yet been reported to date”.

See typos in Figure 2, TraNsfusion

Line 22, Guthrie card

We apologize for that. Typos mistakes has been corrected in Figure 2 and line 200

Line 411-12, This section may be divided by subheadings. It should provide a concise and precise description of the experimental results, their interpretation, as well as the experimental conclusions that can be drawn?

We apologize for the typo in the draft. This comment was left for an editing error in the final version. The sentence has been deleted in the manuscript retyped.

Reviewer 2 Report

This is an article on the nationwide MS/MS-based neonatal screening in Italy. Though there is little of new scientific information included, it will make some contribution to expanding the knowledge on regional epidemiology of the target diseases.

Author Response

Reviewer 2:

This is an article on the nationwide MS/MS-based neonatal screening in Italy. Though there is little of new scientific information included, it will make some contribution to expanding the knowledge on regional epidemiology of the target diseases.

 We thank the reviewer for the good evaluation of the manuscript.
